# Hypernymy Understanding Evaluation of Text-to-Image Models via WordNet Hierarchy

## Abstract

Text-to-image synthesis has recently attracted widespread attention of the community due to rapidly improving generation quality and numerous practical applications. However, little is known about the language understanding capabilities of text-to-image models, making it difficult to reason about prompt formulations that the model would understand well. In this work, we measure the capability of popular text-to-image models to understand *hypernymy*, or the "is-a" relation between words. To this end, we design two automatic metrics based on the WordNet semantic hierarchy and existing image classifiers pretrained on ImageNet. These metrics both enable quantitative comparison of linguistic capabilities for text-to-image models and offer a way of finding qualitative differences, such as words that are unknown to models and thus are difficult for them to draw. We comprehensively evaluate our metrics on various popular text-to-image generation models, including GLIDE, Latent Diffusion, and Stable Diffusion, which allows a better understanding of their shortcomings for downstream applications.

## 1    Introduction

Over the past several years, text-to-image generation has demonstrated remarkable advances (Ramesh et al., 2021; Nichol et al., 2021; Rombach et al., 2022; Ramesh et al., 2022; Saharia et al., 2022) in the quality of generated samples, allowing to create high-fidelity images from a prompt in natural language. These improvements have enabled a variety of practical applications, marking a visible shift in the paradigm of conditional image generation.

Despite the progress in this field, the evaluation of images generated from textual input is still a challenging task. In particular, the majority of works relies on standard metrics for unconditional image generation, such as Frechet Inception Distance (FID, Heusel et al., 2017) on datasets of images paired with their captions, for example, MS-COCO (Lin et al., 2014). As this metric uses captions only as model prompts, it provides an implicit measure of language understanding; similarly, caption-to-image similarity using CLIP (Radford et al., 2021) also does not offer a fine-grained way to understand the language comprehension abilities of the network. However, as correctly visualizing the prompt requires *understanding* the prompt, we are ultimately interested in methods for more in-depth analysis of the model's linguistic competencies.

Several aspects of language understanding are of interest to users of text-to-image generation systems. For example, one crucial aspect is *knowledge of the meaning* of a term: asking a model to draw an object by giving a word that it has not observed during training is unlikely to be successful. Also, if a model is able to draw *only one particular subclass* of an object (for example, only one dog breed when asked to draw a dog) across many samples, it significantly restricts the creative potential of the user for a prompt containing such an object. Even if it is possible to generate an object of another subclass, knowing "difficult categories" for a model in advance can reduce the amount of manual effort and help the user find a model more suitable for their goals.

In this work, we build tools for analyzing the *lexical semantics* capabilities in text-to-image generation models. To construct the metrics for such analysis, we leverage WordNet (Fellbaum, 1998), a well-known lexical database of English words annotated with several semantic relations. Among these relations, we focus on *hypernymy*, or the "is-a" relation. Simply put, hypernymy is the relation between a more general term (for example, "an animal"), called *a hypernym*, and a more specific term (for example, "a dog"), called *a hyponym*.

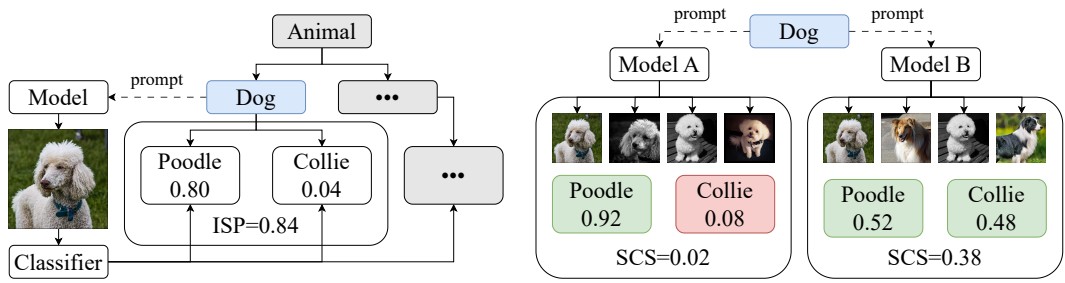

Figure 1: Example calculation of In-Subtree Probability (left) and Subtree Coverage Score (right). Blue color marks the synset used as a prompt.

Using the hypernymy tree from WordNet, we can prompt the model with a specific term (called a *synset*) and measure whether samples of the model with this prompt are in the subtree of the term's hyponyms. Crucially, the WordNet synsets are a superset of classes of ImageNet (Deng et al., 2009), a highly popular dataset for training image classifiers. This correspondence allows us to find the relative positions of concepts depicted by samples and the concept denoted by the prompt using off-the-shelf models pretrained on ImageNet. More specifically, we design two text-to-image generation metrics for scalable measurement of semantics understanding capabilities. The first one, named *In-Subtree Probability* (ISP), shows how well a model generates instances of an object given a specific prompt, while the second one, called *Subtree Coverage Score (SCS)*, displays the coverage of the hyponym subtree for that prompt. Figure 1 contains an ISP and SCS calculation example for a single synset.

We compute ISP and SCS for several popular models, such as GLIDE (Nichol et al., 2021), Latent Diffusion (Rombach et al., 2022), Stable Diffusion, and unCLIP (Ramesh et al., 2022), showing that our metrics generally agree both with existing metrics for text-to-image generation and with human evaluation results. However, the granular nature of our metrics enables a more detailed analysis of linguistic competenices: for example, we show that it is possible to use ISP to find concepts (or meanings of words) unknown to the model. In addition, one can use ISP and SCS to easily compare the performance of two models for a particular set of domains or find domains with the highest disparity between models. We also provide a preliminary analysis of the reasons behind the varying performance of models on different synsets. As we demonstrate, the capability of a model to generate correct hyponyms is connected with the hypernymy knowledge of its language encoder and the frequency of specific synsets in its training data.

In summary, the main contributions of this paper are as follows:

- We propose an evaluation framework for text-to-image generation models that leverages the WordNet hierarchy to assess their hypernymy knowledge. Specifically, we design two interpretable metrics, In-Subtree Probability and Subtree Coverage Score, that measure the generation precision and the coverage of the WordNet tree across different prompts.

- We evaluate a broad range of publicly available models, including Latent Diffusion and Stable Diffusion, using the proposed metrics[1]. We study the influence of the classifier-free guidance scale (Ho & Salimans, 2021), the number of diffusion steps, and the number of generated samples on the behavior of our metrics.

- We demonstrate an example analysis of language understanding capabilities for popular text-to-image models made possible by our evaluation framework. Specifically, we show how to use In-Subtree Probability and Subtree Coverage Score to find concepts that are less known to the model or less diverse in their hyponym distribution.

- We study the connection between the text-to-image model performance (according to ISP and SCS) and the performance of the textual encoder. Furthermore, we compare the per-synset results of text-to-image models and the frequency of objects in standard datasets for training such models, showing that the correlation is higher for weaker models.

---

[1]The code of our experiments is at `github.com/iclr2023-paper/text-to-img-hypernymy`

## 2 BACKGROUND

### 2.1 TEXT-TO-IMAGE GENERATION

Models for generating images from textual prompts have rapidly improved in recent years. Starting from the release of DALL-E (Ramesh et al., 2021) and marked by the emergence of diffusion models (Sohl-Dickstein et al., 2015; Ho et al., 2020), the field has undergone a steady increase in sample fidelity and diversity. Most popular text-to-image models of today, such as Latent Diffusion (LDM, Rombach et al., 2022), Stable Diffusion (SD, Rombach et al., 2022), and Imagen (Saharia et al., 2022), rely on sampling from the reverse diffusion process. The forward diffusion process gradually adds Gaussian noise to images, eventually transforming them into a stationary distribution, and the model learns the reverse process (i.e., generating images from noise) by optimizing a denoising objective. The diffusion process can be controlled with several hyperparameters: the number of diffusion steps, the noise schedule (the magnitude of noise added at each step), and the solver type. These hyperparameters directly affect the quality of samples: for instance, increasing the number of diffusion steps generally results in higher image fidelity (Salimans & Ho, 2022).

Generating images that would correspond to a certain caption is usually done by conditioning diffusion models on the natural language input with a pretrained encoder like BERT (Devlin et al., 2018) or the textual encoder of CLIP (Radford et al., 2021). It is also possible to trade off caption alignment and sample diversity with classifier-free guidance (Ho & Salimans, 2021). This technique blends the conditional and unconditional diffusion processes with weights $w$ and $1 - w$, respectively. Generally, increasing $w$ results in higher similarity between the caption and the image, while decreasing it results in more diverse images.

### 2.2 QUALITY METRICS FOR TEXT-TO-IMAGE SYNTHESIS

The standard practice of the research community is to evaluate text-to-image models in terms of sample quality and the similarity of the image to the prompt. Image quality is usually measured in terms of Inception Score (IS, Salimans et al., 2016b) and Fréchet Inception Distance (FID, Heusel et al., 2017): the first metric uses the outputs of a pretrained ImageNet classifier to estimate the diversity and fidelity of images, while the second metric computes the similarity between representations of model outputs (also extracted from a pretrained model) and representations of a reference image dataset. These metrics assess purely visual aspects of model outputs; by contrast, CLIPScore measures the text-image alignment as the cosine similarity between CLIP embeddings of the prompt and the resulting sample. Although this metric reflects the direct understanding of the prompt, it does not measure the ability of the model to cover the overall visual hierarchy. Moreover, the lack of a predefined hierarchy makes it difficult to derive a holistic proxy measure of model performance across all object categories.

In addition to the above approaches, there exist metrics that target more nuanced skills of text-to-image generation models. Namely, Semantic Object Accuracy (Hinz et al., 2019) measures the ability of a model to depict several objects in the same image using a pretrained object detector. Park et al. (2021) study the ability of text-to-image generators to generalize to novel combinations of objects and their colors or shapes. PaintSkills (Cho et al., 2022) evaluates object recognition, counting, and spatial relation understanding, as well as gender and skin tone biases. Similarly, TISE (Dinh et al., 2022) proposes specific metrics for object fidelity, positional alignment, and counting alignment in text-to-image models. Our work also targets a specific aspect of text-to-image generation; however, unlike the aforementioned studies, we measure more abstract abilities of language understanding *beyond* strict adherence to the input text.

### 2.3 LINGUISTIC CAPABILITIES OF TEXT-TO-IMAGE MODELS

Despite the popularity of models for text-to-image synthesis, the research into their language understanding has mostly been limited to surface-level abilities such as numeracy or compositionality. One particular line of work (Daras & Dimakis, 2022; Millière, 2022; Struppek et al., 2022) examines the sensitivity of text-to-image models to morphology and spelling phenomena such as homoglyphs (pairs of similarly looking symbols). However, to the best of our knowledge, no prior studies have focused on the broader semantic capabilities of such models. Our work addresses this gap by evaluating both overall awareness of the concept hierarchy and the variety of hyponyms for individual concepts.

## 3 METHODOLOGY

This section describes our proposed mechanism for measuring the understanding of hypernymy in text-to-image generation. Specifically, we define the sampling protocol that uses the WordNet database for prompts and introduce two metrics that leverage the structure of WordNet combined with the predictions of ImageNet classifiers for those samples.

### 3.1 OBTAINING SAMPLES USING THE WORDNET TREE

As mentioned in Section 1, we would like to design a metric for hypernymy knowledge of text-to-image models. Hence, we rely on existing annotations for hypernymy in the form of WordNet and map the generated images to nodes in WordNet using pretrained ImageNet classifiers.

However, not all WordNet concepts (grouped into synonym sets or *synsets*) have corresponding classes in the ImageNet dataset, especially in its version with 1,000 classes. Thus, for each class of ImageNet-1k, we take its corresponding synset in the WordNet hierarchy; we call these synsets *leaf nodes*, and we denote the set of leaf nodes as $L$. After obtaining $L$, we take all WordNet synsets that are hypernyms of these leaf nodes and use their union as our evaluation set. For example, for the ImageNet class "green lizard", its hypernyms would include nodes such as "lizard", "reptile", "organism", and "physical entity". Importantly, the leaf nodes themselves are excluded from the evaluation set. We call the set of leaf nodes that can be reached from the synset $s$ its *classifiable subtree*, denoted as $\mathrm{A}(s)$.

Next, we sample a set of images according to the following protocol: for each concept $s$ in the evaluation set, we take its first lemma name and use it as a prompt for a text-to-image model. Each lemma is substituted into the template "An image of a/an lemma." (e.g. "An image of a dog.", "An image of an oven."); in our preliminary experiments, we found that all templates from the set of prompts recommended by Radford et al. (2021) yield similar results. We denote the set of generated images for the synset $s$ as $X_s$. We resize the generated images to $224 \times 224$ using bilinear interpolation to match the input dimensions of ImageNet classifiers.

After we generate samples for each concept, we obtain the class probability distribution $p(y|x)$ for each sample $x$ using a pretrained ImageNet classifier. We then calculate the *hyponym probability distribution* $p_s(y|x)$ for each generated image $x$ of a synset $s$: it is computed as the conditional class distribution given that the generated image is in the classifiable subtree of $s$. More formally,

$$p_s(y|x) = p\left(y|x, y \in A(s)\right), \tag{1}$$

which can be obtained by taking the $\mathrm{softmax}$ of classifier logits over the subset of classes corresponding to the classifiable subtree of $s$. We also define the average distribution of hyponyms $\hat{p}_s(y)$ for the synset $s$ as the following expression:

$$\hat{p}_s(y) = \frac{1}{|X_s|} \sum_{x \in X_s} p_s(y|x). \tag{2}$$

Having computed the probability distribution over hyponyms, we can now design two metrics that leverage this distribution to measure different aspects of hyponymy understanding.

### 3.2 IN-SUBTREE PROBABILITY

First, we would like to measure the correctness of generation: we expect the model to generate less abstract interpretations of the prompt word (i.e., children nodes according to the WordNet hierarchy) and not to generate unrelated concepts. The first metric is called **In-Subtree Probability (ISP)**: we define it as the probability that the generated image lies in the classifiable subtree of the prompt's synset. We average the probabilities over generated images for each synset. More formally,

$$\mathrm{ISP}(s) = \frac{1}{|X_s|} \sum_{x \in X_s} \sum_{c \in A(s)} p(c|x), \tag{3}$$

Naturally, higher values of ISP correspond to outputs that are more consistent with the expectations of the user, and the ideal ISP value is equal to 1.

### 3.3 SUBTREE COVERAGE SCORE

For the second metric, we want to describe the diversity of generated outputs according to the hypernymy relation. Intuitively, we are interested in covering the entire subtree of the synset across many samples while ensuring that each sample represents *only one* object. This prevents two undesirable failure modes: outputs that correspond to "a mixture" of many objects and outputs that cover only one hyponym of the concept. Such properties of unconditional image generators are evaluated by Inception Score (Salimans et al., 2016a), which is why we follow it in the design of our metric named **Subtree Coverage Score (SCS)**. For each concept $s$, we calculate the average Kullback-Leibler divergence between the hyponym probability distribution and the average distribution of hyponyms across all samples generated from $s$ as a prompt:

$$\mathrm{SCS}(s) = \frac{1}{|X_s|} \sum_{x \in X_s} \mathrm{D_{KL}}(p_s(y|x)|\hat{p}_s(y)). \tag{4}$$

As with Inception Score and ISP, the higher the value of SCS, the better. In Appendix H, we compare Subtree Coverage Score with a simpler diversity metric that uses the entropy of classifier predictions, finding that SCS better aligns with human preferences.

### 3.4 AGGREGATING RESULTS

Each of the above metrics measures the results for a single synset. To get the final metric value for a single model, we average the metrics across all synsets from the evaluation set and divide the result by the maximum possible value (1.0 for ISP and $\approx 1.624$ for SCS) for ease of interpretation. One may also note that $\mathrm{SCS}(s)$ is always equal to 0 when $s$ has only one node in $\mathrm{A}(s)$, as it reduces to the average of Kullback-Leibler divergences for identical distributions. Therefore, we exclude these synsets from aggregation in the case of Subtree Coverage Score; however, we keep them when calculating the model's In-Subtree Probability.

This direct averaging treats all synsets equally regardless of their position in the WordNet hierarchy, causing the metrics to be incomparable between synsets from different levels. Indeed, higher nodes have more hyponyms by construction: for instance, the value of ISP for "entity" (the root of the WordNet tree) is always equal to 1. As a result, values from different levels of WordNet might skew the aggregated metric. Future work might address this issue, for example, by applying a discounting factor to higher levels of the hierarchy. However, in this paper, we aim to introduce the approach of hierarchical evaluation and thus leave this question out of the scope of our study.

## 4 EXPERIMENTS

In this section, we evaluate several popular text-to-image models with ISP and SCS to compare our metrics with other approaches, including human evaluation. We also study the influence of several generation hyperparameters on the behavior of the proposed metrics.

### 4.1 SETUP

We run the experiments on the following text-to-image models: GLIDE (Nichol et al., 2021), Latent Diffusion (Rombach et al., 2022), Stable Diffusion 1.4[2], Stable Diffusion v2[3], unCLIP (Ramesh et al., 2022), Kandinsky 2.1 (Razzhigaev et al., 2023), DeepFloyd IF (DeepFloyd Lab, 2023), and Stable Diffusion XL (Podell et al., 2023). We use an open-source version of unCLIP (Lee et al., 2022) in our experiments, as the original one is not publicly available. We chose these models because they are openly available and were close to state-of-the-art at the moment of their release. We use ViT-B/16 (Dosovitskiy et al., 2020) as the ImageNet classifier due to its high accuracy, low calibration error, and high robustness (Naseer et al., 2021). We experimented with different pretrained classifiers and found that they resulted in highly similar rankings for both metrics: more details on the choice of the classifier are available in Appendix A.

We generate 32 images for each synset using the default DDIM sampler with $\eta = 0$: experiments with other numbers of samples can be seen in Appendix B. We run each model in 16-bit precision to

---

[2]`huggingface.co/CompVis/stable-diffusion-v1-4`
[3]`huggingface.co/stabilityai/stable-diffusion-2-base`

Table 1: Model performance according to ISP, SCS, and baseline metrics. The best values are in bold.

| Model | Precision | | Diversity | |
|---|---|---|---|---|
| | ISP ↑ | CLIPScore ↑ | SCS ↑ | FID ↓ |
| GLIDE | 0.221 | 0.279 | 0.198 | 37.93 |
| Latent Diffusion | 0.217 | 0.304 | 0.182 | 36.43 |
| Stable Diffusion 1.4 | 0.329 | 0.314 | **0.256** | 16.57 |
| Stable Diffusion 2.0 | 0.297 | 0.317 | 0.233 | 16.25 |
| unCLIP | 0.352 | 0.322 | 0.194 | 18.29 |
| Kandinsky 2.1 | 0.345 | 0.322 | 0.164 | 18.97 |
| DeepFloyd IF XL | **0.357** | 0.323 | 0.158 | 21.48 |
| Stable Diffusion XL | 0.345 | **0.324** | 0.196 | **15.03** |

speed up the generation process. We use 50 base model steps with 27 upsampler steps for GLIDE, 50 diffusion steps for Latent Diffusion and all Stable Diffusion models, and 25 prior, 25 decoder and 7 super-resolution steps for unCLIP: Section 4.4 describes our experiments with other numbers of steps. We set the classifier-free guidance (Ho & Salimans, 2021) weight to 7.5 in all experiments unless stated otherwise.

## 4.2 RESULTS

First, we compare the models using the metrics proposed in Section 3, along with Fréchet Inception Distance (Heusel et al., 2017) and CLIPScore (Hessel et al., 2021) as baselines. This comparison is intended to be a form of a "sanity check" for ISP and SCS: one would expect that models generally viewed as better generators would also be better at hypernymy knowledge. FID and CLIP are computed on 10,000 random prompts from the MS-COCO Lin et al. (2014) $512 \times 512$ validation set. We present the results of the experiment in Table 1: importantly, the ranking of models is mostly consistent within metrics of similar categories. Both ISP and SCS have the relative standard deviation of less than $1\%$ when computed over four random seeds.

## 4.3 HUMAN EVALUATION

In this experiment, we measure the correlation of In-Subtree Probability and Subtree Coverage Score with the human understanding of hyponymy. To do this, we conduct crowdsourced evaluations of text-caption similarity and sample diversity for several text-to-image models. To estimate text to caption similarity, we present the annotators with two generated images along with the caption from which they were generated. The workers are then tasked to select the image that best matches the text description. For the diversity evaluation, we show the annotators two collections of generated images and ask them to select the grid with more diverse samples.

The models are evaluated on a random subset of 20 synsets from the WordNet hierarchy; we generate 20 pairs of images (or grids) per concept, which results in 400 tasks per comparison with the overlap of 5 labelers. We also report Krippendorff's alpha (Krippendorff, 2018) as a measure of inter-annotator agreement. Further details of the human evaluation protocol, including the annotation interface, are shown in Appendix C.

We compare Stable Diffusion 1.4 with clasifier-free guidance of 7.5 against Latent Diffusion, unCLIP, and Stable Diffusion 1.4 that has a lower guidance value of 2.5. The results of this evaluation can be seen in Table 2: in general, the differences in all metrics follow human preferences.

Table 2: Results of human preference evaluation for models compared with Stable Diffusion 1.4. Krippendorff's alpha is in subscript.

| Model | Caption similarity | | | Sample diversity | | |
|---|---|---|---|---|---|---|
| | Human ↑ | ΔISP ↑ | ΔCLIPScore ↑ | Human ↑ | ΔSCS ↑ | ΔFID ↓ |
| Latent Diffusion | 17.1% $_{0.75}$ | -0.112 | -0.010 | 21.9% $_{0.58}$ | -0.074 | 19.86 |
| unCLIP | 49.1% $_{0.82}$ | 0.023 | 0.080 | 26.8% $_{0.63}$ | -0.062 | 1.72 |
| SD 1.4 ($w = 2.5$) | 25.3% $_{0.81}$ | -0.060 | -0.080 | 57.5% $_{0.58}$ | 0.033 | -5.09 |

Table 3: Synset-level Spearman rank correlations of metric differences and human preferences. The subscript shows p-values for correlations. The best values in each category are in bold.

| Model | Caption similarity | | Sample diversity | |
|---|---|---|---|---|
| | ISP $\uparrow$ | CLIPScore $\uparrow$ | SCS $\uparrow$ | Inception Score $\uparrow$ |
| Latent Diffusion | **0.41** $_{0.00}$ | -0.63 $_{0.00}$ | **0.52** $_{0.00}$ | 0.33 $_{0.04}$ |
| unCLIP | **0.63** $_{0.00}$ | -0.10 $_{0.53}$ | **0.44** $_{0.00}$ | 0.38 $_{0.02}$ |
| SD ($w = 2.5$) | **0.63** $_{0.00}$ | 0.59 $_{0.00}$ | **0.40** $_{0.01}$ | 0.38 $_{0.02}$ |

Next, we compute rank correlations between synset metric differences and annotator preferences in Table 3 to measure detailed agreement. Unlike CLIPScore and Inception Score (used here due to a lack of references for FID), both ISP and SCS have moderate yet statistically significant correlation with human preference and thus are better for granular evaluation.

## 4.4 IMPACT OF THE NUMBER OF DIFFUSION STEPS

When evaluating machine learning models, one needs to balance the metric computation time and the accuracy of measurement. In case of diffusion models, this can be easily done by adjusting the number of steps in the reverse diffusion process: fewer steps generally lead to lower image quality. In this experiment, we aim to determine the optimal number of steps that would be necessary for ISP and SCS. Specifically, we compute these two metrics on Latent Diffusion and Stable Diffusion v1.4 with the number of steps $T$ from the following set: $\{5, 10, 15, 25, 50, 75, 100\}$.

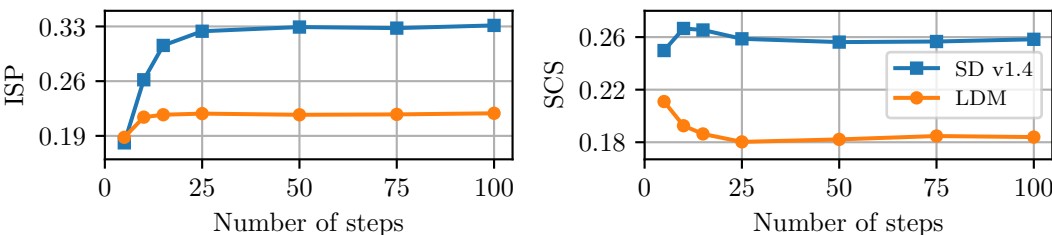

Figure 2: ISP and SCS values depending on the number of diffusion steps.

Figure 2 displays the outcome of this experiment: we find that both ISP and SCS are unstable when the number of steps is less than 25, which is expected, because the quality of images deteriorates when $T$ is too low (Salimans & Ho, 2022). Increasing the number of diffusion steps beyond this point, however, has little to no effect on the results. We also note that, unlike In-Subtree Probability, Subtree Coverage Score increases at small values of $T$. We attribute this to the fact that SCS measures the diversity of classifier predictions, which might be high for out-of-distribution inputs or images with excessive noise.

## 4.5 IMPACT OF CLASSIFIER-FREE GUIDANCE

As we discussed in Section 2, classifier-free guidance is a technique that allows to trade off sample precision for diversity. To study the influence of the guidance weight on our metrics, we repeat the experiments of Section 4.2 for all models using the $w$ values of $\{1.0, 1.5, 2.0, 2.5, 5.0, 7.5, 10.0\}$.

Our findings are shown in Figure 3: as anticipated, higher guidance leads to better precision (indicated by higher ISP) and lower guidance leads to more diverse samples (as indicated by higher SCS). We note that excessively high or low guidance values may result in both lower SCS and lower ISP, which hints at the presence of generation artifacts. We also observe that the ranking of models in Table 1 agrees with the relative positions of Pareto frontiers obtained in this experiment: this means that it is possible to use our metrics with different guidance scales depending on the application and expect similar results. Intuitively, hypernymy knowledge is a skill that is independent of high-fidelity image generation ability, which is consistent with the results we obtain here.

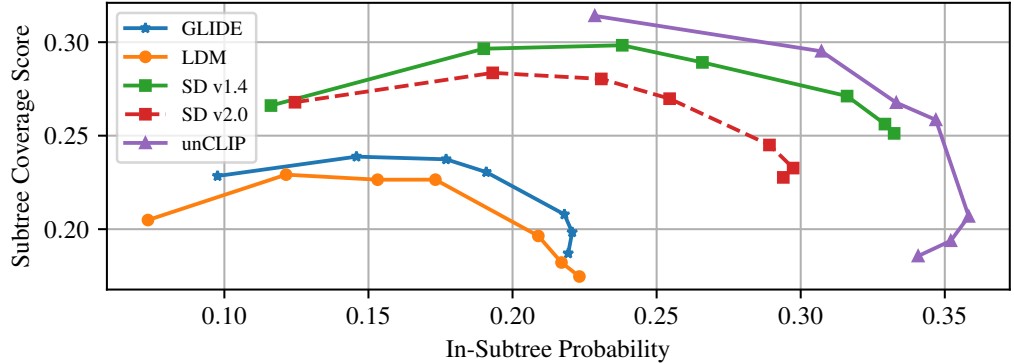

Figure 3: Results of evaluation with different guidance scales.

# 5 ANALYSIS

## 5.1 FINDING UNKNOWN CONCEPTS

Using In-Subtree Probability, it is easy to determine which concepts are drawn poorly by the model by taking synsets with low values of this metric. To demonstrate this use case, we select synsets that are among the lowest ones in terms of ISP across different models. More illustrative synsets are displayed in Figure 4, and a random selection of synsets is presented in Figure 11 of Appendix D.

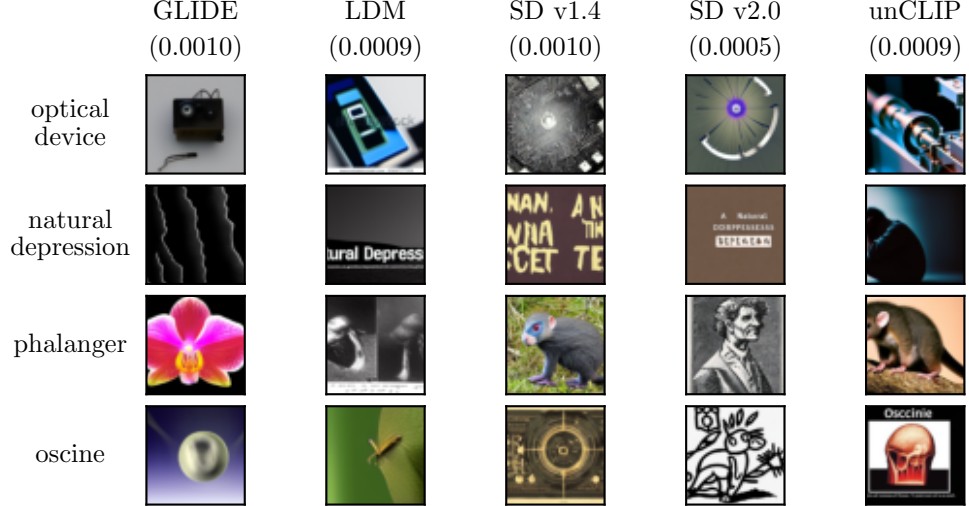

Figure 4: Outputs for synsets with low ISP. Average model ISP for these synsets is in parentheses.

As we can see, our approach not only uncovers inherently unknown concepts (such as "phalanger" or "oscine"), but also detects homonyms for which the models are only familiar with one meaning (e.g., "convertible" or "landing"). Additionally, it identifies synsets where the models only recognize some of its hyponyms (e.g., "contestant"). In some cases, the model generates a coherent output, but the concept understanding is insufficient to achieve high ISP (e.g., "optical device"). We perform an identical analysis with Subtree Coverage Score to find concepts with low diversity in Appendix E.

## 5.2 GRANULAR COMPARISON OF MODELS

We can also compare two models in terms of how well they generate individual concepts. To do this, we calculate the differences between ISP and SCS for each synset and rank synsets according to the resulting differences. We present this analysis for Stable Diffusion 1.4 and Stable Diffusion 2.0 in Figure 5. Such comparison allows us to more easily understand the relative strengths and weaknesses of each model with direct illustrations. For instance, we can see that the models are almost always equal, and yet still have synsets with drastic metric differences. Apart from analyzing model performance on specific concepts, it is also possible to evaluate them on entire synset subtrees, which we discuss in Appendix F.

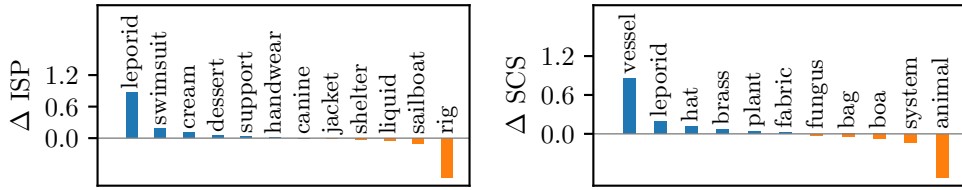

Figure 5: Per-synset comparison between Stable Diffusion v1.4 and Stable Diffusion v2.0. The vertical axis denotes the differences of metrics between the former and the latter model.

## 5.3 RELATIONSHIP WITH TRAINING DATA

We hypothesize that poor representation of some concepts may depend on their frequency in the training corpus. We analyze three popular multimodal datasets, LAION-400M (Schuhmann et al., 2021), LAION-2B-en (Schuhmann et al., 2022) and COYO (Byeon et al., 2022), counting the number of times that each WordNet concept appeared in the text captions. These three datasets have significant presence in the training data of the models we use: Latent Diffusion was trained on LAION-400M, Stable Diffusion v1.4 was trained on LAION-2B-en and then finetuned, Stable Diffusion v2.0 was trained on a superset of LAION-2B-en and then finetuned, and the unCLIP variation we used was partially trained on COYO. After computing the frequencies, we measure the Spearman rank correlation between the synset counts and the per-synset metrics of the models we evaluate in our primary experiments.

Table 4: Spearman rank correlation between synset metrics and their frequency in the dataset. P-values are in subscript, statistically significant results ($p < 0.05$) are in bold.

| Model | In-subtree Probability | | | Subtree Coverage Score | | |
|---|---|---|---|---|---|---|
| | LAION-400M | LAION-2B | COYO | LAION-400M | LAION-2B | COYO |
| GLIDE | $\mathbf{0.19}_{0.00}$ | $\mathbf{0.18}_{0.00}$ | $\mathbf{0.16}_{0.00}$ | $\mathbf{0.28}_{0.00}$ | $\mathbf{0.29}_{0.00}$ | $\mathbf{0.29}_{0.00}$ |
| LDM | $\mathbf{0.29}_{0.00}$ | $\mathbf{0.27}_{0.00}$ | $\mathbf{0.24}_{0.00}$ | $\mathbf{0.15}_{0.00}$ | $\mathbf{0.16}_{0.00}$ | $\mathbf{0.17}_{0.00}$ |
| SD v1.4 | $0.06_{0.15}$ | $0.04_{0.34}$ | $0.01_{0.81}$ | $0.00_{0.15}$ | $0.01_{0.34}$ | $0.03_{0.81}$ |
| SD v2.0 | $\mathbf{0.10}_{0.01}$ | $\mathbf{0.08}_{0.04}$ | $0.05_{0.18}$ | $\mathbf{0.07}_{0.01}$ | $\mathbf{0.08}_{0.04}$ | $0.08_{0.18}$ |
| unCLIP | $0.02_{0.63}$ | $0.00_{0.91}$ | $-0.02_{0.61}$ | $0.04_{0.63}$ | $0.05_{0.91}$ | $0.08_{0.61}$ |

As we can see from Table 4, the majority of correlations are not high in magnitude yet still significant, which suggests that hypernymy understanding and concept knowledge cannot be attributed purely to the frequency of specific synsets in training data. Weaker models also tend to have higher correlations, whereas the results for stronger models are less pronounced. This difference might arise due to the finetuning procedures on aesthetic images or simply higher capacity of better models. Alternatively, the hyponymy performance of text-to-image models might arise purely from the semantic capabilities of the part of the model that encodes the prompt. In Appendix G, we provide results of evaluation for the CLIP language encoder, showing that there is a high and significant correlation between average hyponym embedding similarities and metric values for a given synset.

## 6 CONCLUSION

In this work, we introduce In-Subtree Probability and Subtree Coverage Score, two metrics for evaluating the language understanding capabilities of text-to-image models. We validate these metrics by comparing them to standard evaluation methods and human judgment. Through extensive analysis, we demonstrate how ISP and SCS can provide a deeper understanding of text-to-image models and their semantic abilities.

Future work might address the limitation of our approach connected to its reliance on WordNet and ImageNet: these datasets do not contain the entire concept hierarchy, and therefore it might be valuable to study data-driven hierarchies (such as the ones proposed by Desai et al., 2023) based on the actual use cases of text-to-image models. Furthermore, models that explicitly leverage ImageNet data (such as all models using the CLIP encoder) might have an unfair advantage due to a smaller domain shift and thus obtain inflated ISP and SCS scores.

ETHICS STATEMENT

Text-to-image models trained on large-scale web data are able to generate sensitive or offensive content. We do not directly improve these capabilities of the models and, instead, offer a way to more thoroughly monitor their performance, which could help decrease undesired behavior. We use human annotators as part of our research. The workers were paid above the minimum wage in their respective countries, please see Appendix C for details.

REPRODUCIBILITY STATEMENT

Our work makes the following efforts to ensure reproducibility: we release the code for our experiments and analyses, we describe the setup of our experiments and hyperparameter choices in Section 4.1, and we provide details on the human evaluation protocol in Section 4.3 and Appendix C.

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

## A  CLASSIFIER CHOICE

Because our metrics fundamentally depend on the quality of the pretrained clasifier, we investigate how choosing different classifiers impacts our results. Specifically, we compute ISP and SCS with three different classifiers: ViT-B/16 (Dosovitskiy et al., 2020), ConvNeXt-B (Liu et al., 2022) and ResNet-50 (He et al., 2016). The results can be seen in Table 5: importantly, the values of synset metrics have significant pairwise rank correlations for each specific model (see Table 6). We conclude that while the exact values of ISP and SCS can differ significantly, all classifiers rank the models (along with synsets within one model) in a similar way.

Table 5: Comparison of metric values for different classifiers.

| Model | ISP ↑ | | | SCS ↑ | | |
|---|---|---|---|---|---|---|
| | ViT-B/16 | ConvNeXt-B | ResNet-50 | ViT-B/16 | ConvNeXt-B | ResNet-50 |
| GLIDE | 0.221 | 0.188 | 0.220 | 0.198 | 0.180 | 0.243 |
| LDM | 0.218 | 0.190 | 0.218 | 0.180 | 0.161 | 0.218 |
| SD v1.4 | 0.329 | 0.277 | 0.349 | **0.258** | **0.221** | **0.272** |
| SD v2.0 | 0.296 | 0.254 | 0.307 | 0.232 | 0.205 | 0.259 |
| unCLIP | **0.351** | **0.299** | **0.363** | 0.190 | 0.157 | 0.211 |

Table 6: Mean pairwise Spearman rank correlation between synset metrics for three classifiers. All results are statistically significant ($p < 0.05$).

| Model | ISP | SCS |
|---|---|---|
| GLIDE | 0.98 | 0.89 |
| LDM | 0.97 | 0.88 |
| SD v1.4 | 0.98 | 0.91 |
| SD v2.0 | 0.98 | 0.91 |
| unCLIP | 0.97 | 0.89 |

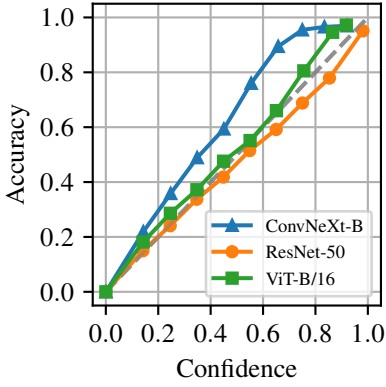

Figure 6: Calibration curves on the ImageNet validation set.

To select the best classifier, we compute the expected calibration error (ECE) and accuracy on the ImageNet validation set for the three candidates and report the results in Table 7. We also plot the calibration curves of all models in Figure 6. Notably, while ConvNeXt-B has the highest accuracy, it is the most miscalibrated model, and therefore we use ViT-B/16 as the classifier for our metrics.

Table 7: Expected calibration error (ECE) and accuracy for the ImageNet validation set. ECE is computed using 100 bins.

| Classifier | ECE ↓ | Accuracy ↑ |
|---|---|---|
| ViT-B/16 | **0.035** | 0.81 |
| ConvNeXt-B | 0.133 | **0.84** |
| ResNet-50 | 0.036 | 0.76 |

# B  METRIC STABILITY

Because our approach fundamentally depends on the number of generated samples per synset, we investigate how changing this value affects the final metrics. Specifically, we conduct four separate runs for the number of samples from 4 to 32 and measure the average metric values, as well as their standard deviations. Figure 7 shows the results of this experiment: we find that both metrics are stable across the analyzed setups with standard deviation rarely exceeding 1% of the average value. We also note that Subtree Coverage Score increases with the number of samples, which is expected for a diversity measure.

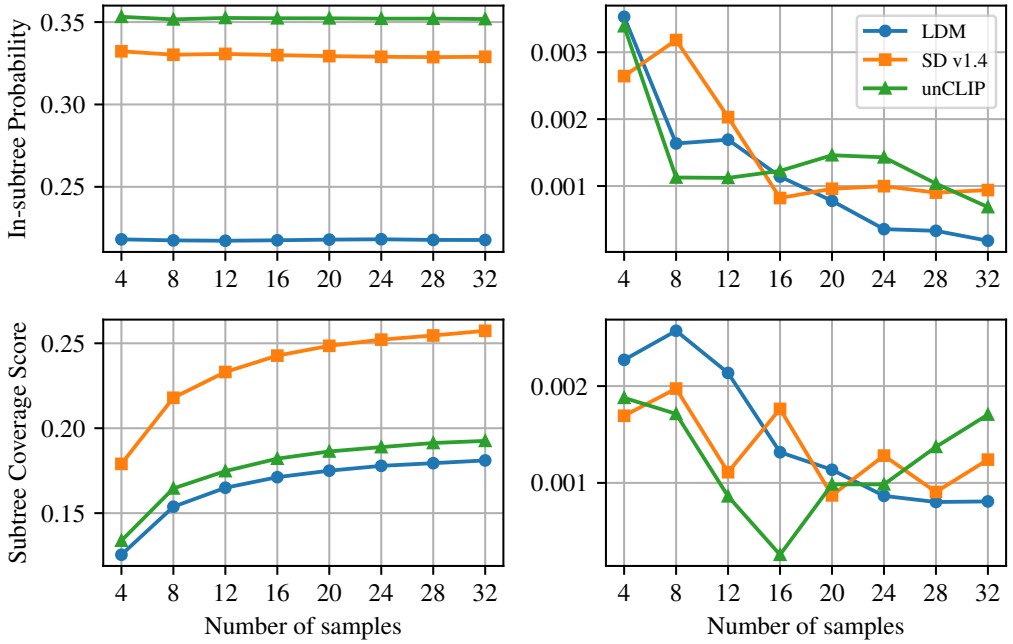

Figure 7: Metric values (left) and standard deviations (right) as a function of the number of generated samples in each synset.

In addition, we analyze how different seeds impact the ranking of synset ISP and SCS within one model in Table 8. We discover that different runs have high pairwise correlations ($> 0.97$ on average for ISP and $> 0.94$ for SCS) and conclude that the metrics are also stable on a per-synset level.

Table 8: Mean pairwise Spearman's rank correlation between synset metrics for four separate seeds. All results are statistically significant with $p < 0.05$.

| Model | ISP | SCS |
|---|---|---|
| GLIDE | 0.975 | 0.955 |
| Latent Diffusion | 0.980 | 0.943 |
| Stable Diffusion 1.4 | 0.983 | 0.947 |
| Stable Diffusion 2.0 | 0.984 | 0.954 |
| unCLIP | 0.989 | 0.944 |

# C  HUMAN EVALUATION DETAILS

Our evaluations were conducted on samples from the following 20 synsets: *frog, clock, oven, monkey, knife, wolf, pan, boat, wheel, shark, whale, fruit, turtle, hat, vegetable, pot, flower, duck, chair, spider*. The synsets were chosen randomly among those with a distance to the closest leaf node no greater than 2: this was done to eliminate overly abstract concepts for ease of interpretation by crowd workers. We manually discarded words with different possible meanings, such as "rail".

Table 9: In-subtree probability for different subtrees of the ImageNet hierarchy. The highest value is emboldened, and the second highest is underlined.

| Subset | GLIDE | LDM | SD v1.4 | SD v2.0 | unCLIP | Kandinsky 2.1 | IF XL | SD XL |
|---|---|---|---|---|---|---|---|---|
| Vessel | 0.282 | 0.497 | 0.512 | 0.578 | **0.584** | 0.532 | 0.513 | 0.541 |
| Furniture | 0.267 | 0.333 | 0.481 | 0.384 | 0.436 | 0.403 | 0.508 | **0.516** |
| Bird | 0.470 | 0.271 | 0.462 | 0.420 | 0.425 | 0.431 | 0.499 | **0.531** |
| Clothing | 0.065 | 0.206 | 0.247 | 0.172 | 0.276 | 0.230 | **0.296** | 0.242 |
| Lizard | 0.346 | 0.175 | 0.289 | 0.263 | 0.295 | 0.296 | 0.294 | **0.401** |
| Fruit | **0.492** | 0.374 | 0.438 | 0.329 | 0.452 | 0.469 | 0.416 | 0.275 |
| Full tree | 0.221 | 0.218 | 0.329 | 0.296 | 0.351 | 0.345 | **0.357** | 0.345 |

We provide task descriptions in Figures 8 and 9, and the evaluation interface is shown in Figure 10. The participants were paid \$0.10 per one task, which is above the hourly minimum wage in their geographical regions. We required that participants complete 5 manually labeled training tasks and achieve an accuracy of more than 60% on them before starting the evaluation procedure.

The tasks were presented in groups of five. We included one control task in each group to filter automatically generated responses. For text to caption similarity, the control tasks had one regular image and one image generated from a different synset. For image diversity, control tasks had one normal grid and one grid that consisted of four identical images. Participants who failed two control tasks in a row were banned. We also included measures against fast responses.

## D  ADDITIONAL SYNSETS WITH LOW IN-SUBTREE PROBABILITY

In Figure 11, we present randomly sampled concepts that were among the lowest 50 in terms of average model ISP.

## E  FINDING CONCEPTS WITH LOW DIVERSITY

Similarly to the analysis of Section 5.1, it is also possible to find concepts that have low diversity for the given model. Here we analyze Stable Diffusion 1.4 by selecting random concepts that have low Subtree Coverage Score and plot them in 12. Our findings are highly interpretable: for example, "belgian sheepdog" has four varieties: "groenendael", "malinois", "tervuren" and "laekenois", and only the first two are parts of the ImageNet hierarchy. The model only draws the "groenendael", and thus the coverage score is very low.

## F  SUBTREE COMPARISON

Our approach makes it easy to evaluate models on a particular set of concepts by simply averaging synset metrics over it. The hierarchical nature of the ImageNet tree also simplifies the process of finding large sets of semantically connected words: one can simply take entire hyponym subtrees of concepts of interest. We compare a wide range of models on an illustrative set of concept subtrees in Tables 9 and 10. Notably, model rankings on these sets significantly differ from metrics computed over the entire hierarchy. This highlights the advantage of our method: we are able to go beyond what a single metric value would give us.

## G  RELATIONSHIP WITH THE TEXTUAL ENCODER

Because the metric values for models vary across synsets, a natural question is whether the quality for a given concept corresponds to the knowledge about this concept contained in the textual encoder of the model. To verify this, we conduct a comparison of performance across synsets with the similarity of each synset to its hyponyms, using the values of ISP and SCS for Stable Diffusion v1.4, which uses CLIP ViT-L/14 text encoder for conditioning on its prompts.

Table 10: Subtree Coverage Score for different subtrees of the ImageNet hierarchy. The highest value is emboldened, and the second highest is underlined.

| Subset | GLIDE | LDM | SD v1.4 | SD v2.0 | unCLIP | Kandinsky 2.1 | IF XL | SD XL |
|--------|-------|-----|---------|---------|--------|---------------|-------|-------|
| Vessel | 0.188 | 0.183 | **0.267** | 0.205 | 0.187 | 0.115 | 0.077 | 0.150 |
| Furniture | **0.211** | 0.167 | 0.190 | 0.182 | 0.183 | 0.128 | 0.136 | 0.115 |
| Bird | 0.152 | 0.137 | **0.168** | 0.160 | 0.092 | 0.115 | 0.107 | 0.120 |
| Clothing | 0.090 | 0.160 | **0.193** | 0.148 | 0.139 | 0.119 | 0.114 | 0.155 |
| Lizard | 0.084 | 0.104 | 0.064 | **0.119** | 0.086 | 0.059 | 0.061 | 0.061 |
| Fruit | 0.203 | 0.158 | 0.218 | **0.240** | 0.205 | 0.146 | 0.119 | 0.139 |
| Full Tree | 0.198 | 0.180 | **0.258** | 0.232 | 0.190 | 0.164 | 0.158 | 0.196 |

More specifically, for each synset from the evaluation set, we obtain the CLIP text encoder embeddings for this synset, as well as the embeddings for all its hyponyms contained in the set of ImageNet classes. We exclude all other hyponyms to ensure a proper comparison with the ISP and SCS. After that, we compute the average cosine similarity of each synset to its hyponyms and compute the correlation of these similarities to ISP and SCS across a range of classifier-free guidance values.

Table 11: Spearman correlation of CLIP hyponym similarities with WordNet-based metrics for Stable Diffusion 1.4. All results are statistically significant ($p < 0.05$).

| Guidance | In-Subtree Probability | Subtree Coverage Score |
|----------|------------------------|------------------------|
| 2.5 | 0.397 | -0.139 |
| 5.0 | 0.405 | -0.178 |
| 7.5 | 0.400 | -0.186 |
| 10.0 | 0.393 | -0.192 |

The results of this evaluation are available in Table 11. As we can see, the cosine similarity of synsets to their hyponyms significantly correlates with the In-Subtree Probability, which suggests a connection between the knowledge of the hypernymy relationship of the encoder and the performance of the full model according to this metric. On the other hand, Subtree Coverage Score displays a negative correlation, which might be caused by more diverse subtrees with inaccurate representation of the prompt having higher scores.

## H  COMPARISON OF SUBTREE COVERAGE SCORE AND AVERAGE ENTROPY

One common approach to estimating diversity is computing the entropy of class distribution. However, it heavily relies on the assumption that all objects have distinct classes assigned to them. This is not the case in our setup, because we utilize a pretrained classifier which may give any distribution as output for a single object. If all generated images are nearly identical and they represent some mixture between subtypes (for example, a blend between all dog breeds), the entropy of the average distribution $\mathcal{H}(\hat{p}_s)$ is going to be extremely high, while the actual diversity is low. Therefore, we penalize our diversity measure when the predicted objects do not belong to distinct classes by using the Inception Score formulation. In Table 12, we compare the correlation of human preference to Subtree Coverage Score and average entropy, similarly to the setup of Section 4.3. We find that SCS has a higher correlation and that Average Entropy does not provide statistically significant results for unCLIP and Latent Diffusion, which supports our choice of metric inspired by the Inception Score.

| Model | Subtree Coverage Score ↑ | Average Entropy ↑ |
|---|---|---|
| Latent Diffusion | $\mathbf{0.52}_{0.00}$ | $0.26_{0.02}$ |
| unCLIP | $\mathbf{0.44}_{0.00}$ | $0.21_{0.18}$ |
| SD ($w = 2.5$) | $\mathbf{0.40}_{0.01}$ | $0.31_{0.05}$ |

Table 12: Synset-level Spearman rank correlations of metric differences and human preferences. The subscript shows p-values for correlations. The best values in each category are in bold.

Two neural networks tried to generate an image of an object given the text caption.

Please help us understand which image better matches the object given in the text.

**How to answer the question:**

For all comparisons we provide a text description from which these images were created. Text description contains a reference to some object (e.g. "An image of a cat"). To answer the question, we suggest using the following algorithm. For each generated image:

- First, read the text description (e.g. "An image of a cat").
- If no images correspond to the object select option "equal".
- If only one image corresponds to the object and another one does not: select the image that corresponds to the text.
- If both images correspond to the text: it is up to you whether to select "equal" or to choose the one that in your opinion corresponds to the text more precisely.

Figure 8: Text to caption similarity task description.

Two neural networks tried to generate an image of an object given the text caption. We present to you two grids of 4 generated images each.

Please help us understand which grid of images is more diverse.

**What do we mean by diversity:**

A grid is diverse if it has variation in the generated object. Some examples of variation include:
- Different animal species (e.g. a persian cat and a sphinx cat).
- Different subtypes of an object: (e.g. a race car, a sedan car).
- Different colors: (e.g. a black cat and a white cat).
- Different positions of the same object (e.g. a running human and a sitting human).
- Different details on the same object (e.g. a human wearing glasses and a human wearing a monocle).

**How to answer the question:**

To answer the question, we suggest using the following algorithm. For each pair of grids:
- If only one grid has diverse images, and the other one has little variation: select the grid that is diverse.
- If none of the grids has diverse images, and both of them have little variation: select "equal".
- If both images have some level of diversity, it's up to you whether to select: "equal" or to choose the one that in your opinion has more diversity.

Figure 9: Diversity task description.

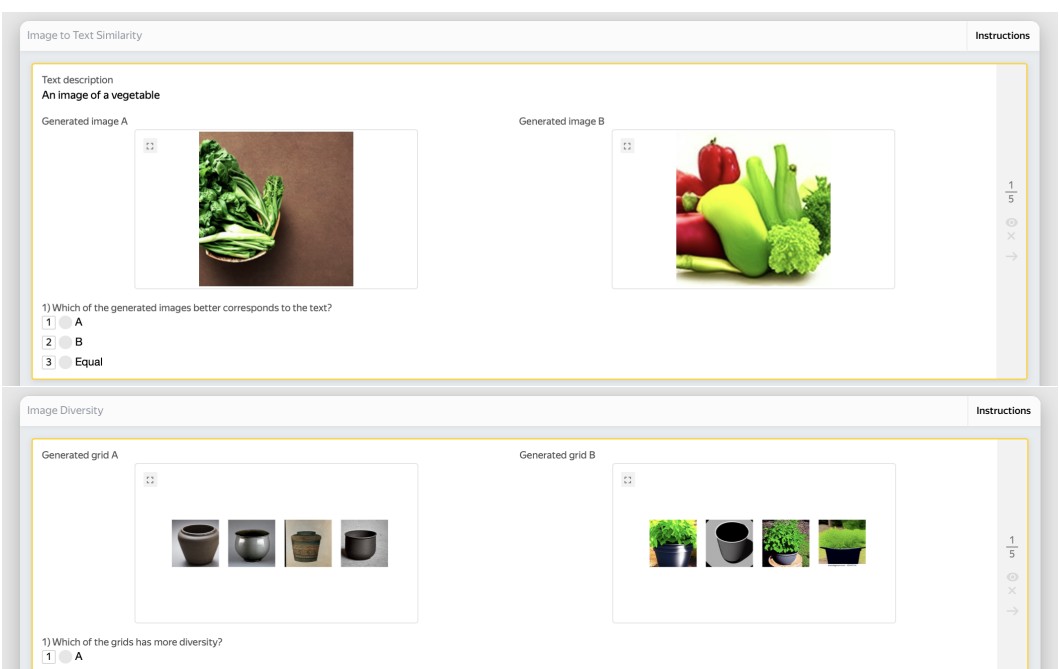

Figure 10: Screenshots from the evaluation interface. Top is caption similarity, bottom is diversity.

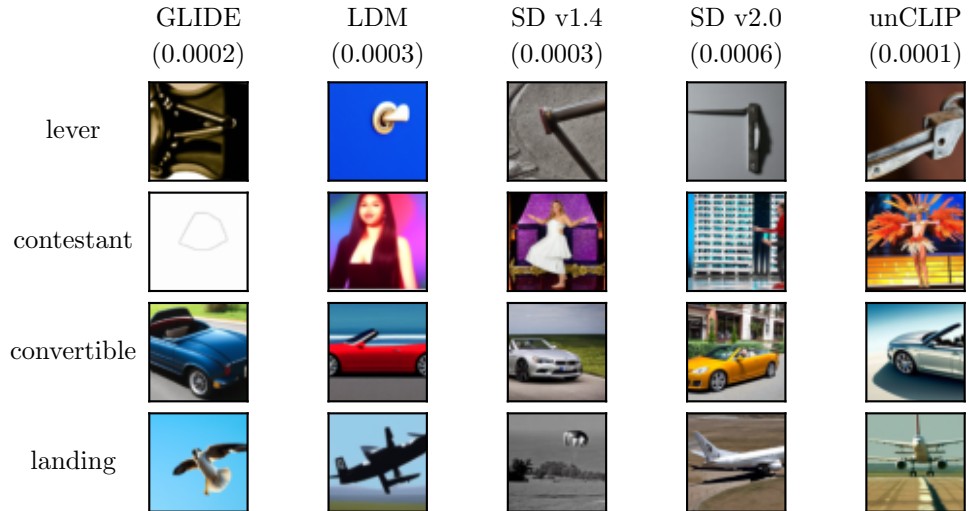

Figure 11: Generated images of randomly selected synsets with low ISP. Average model ISP for these synsets is presented in parenthesis.

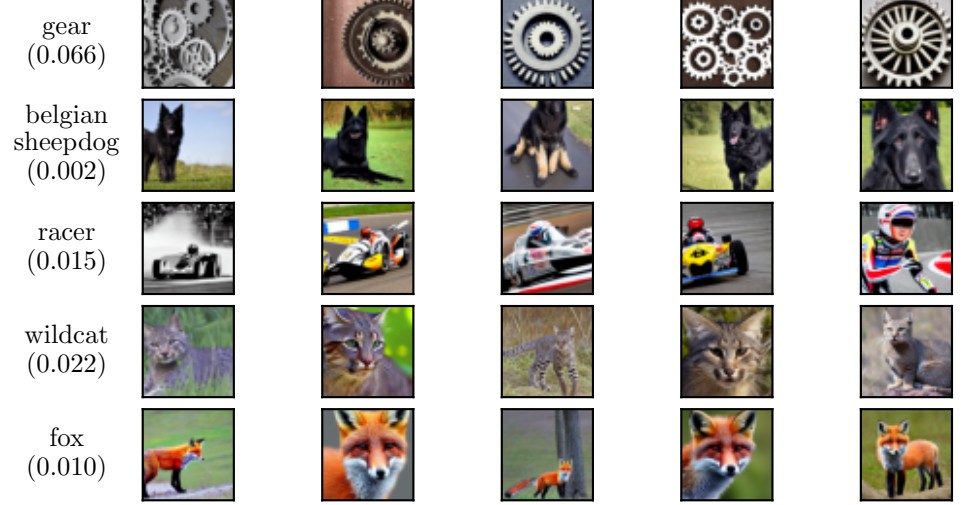

Figure 12: Generated images of randomly selected synsets with low SCS for Stable Diffusion 1.4. Synset SCS is presented in parenthesis.

