# OpenReview forum: "Hypernymy Understanding Evaluation of Text-to-Image Models via WordNet Hierarchy"
_ICLR.cc/2024/Conference — Submitted to ICLR 2024_

### Official Review · Reviewer_4BDw · 2023-10-24

**Soundness:** 3 good
**Presentation:** 2 fair
**Contribution:** 3 good
**Rating:** 6
**Confidence:** 4

**Summary:**

This paper proposes a framework to evaluate the hypernymy understanding of text-to-image models using the WordNet hierarchy and ImageNet classifiers. The paper introduces two metrics, In-Subtree Probability (ISP) and Subtree Coverage Score (SCS), that measure the generation precision and coverage of the WordNet tree. The paper also compares several popular text-to-image models, such as GLIDE, Latent Diffusion, and Stable Diffusion, using the proposed metrics and analyzes their language understanding capabilities and limitations.

**Strengths:**

The paper introduces a novel framework for evaluating text-to-image models using the WordNet hierarchy and ImageNet classifiers, which is a unique approach in the field. The work is of high quality, with rigorous methodology and comprehensive evaluation of several popular models. The research is significant as it provides valuable insights into the language understanding capabilities of text-to-image models, which can guide future research in this area.

**Weaknesses:**

The paper could include more diverse models for comparison to provide a more comprehensive evaluation. Meanwhile, the proposed metrics (ISP and SCS) are innovative, but their interpretation and implications could be explained more clearly.

**Questions:**

NA

---

> ### Author Response · Authors · 2023-11-20
> **Author Response to Official Review by Reviewer 4BDw**
>
> Thank you for your review! Below, you can find our responses to your comments:
>
> > The paper could include more diverse models for comparison to provide a more comprehensive evaluation.
>
> Thank you for the suggestion! We included **3 more models** (currently considered state-of-the-art among openly available ones) in the new revision of the paper; please see the **updated PDF and the general response** for details.
>
> > the proposed metrics (ISP and SCS) are innovative, but their interpretation and implications could be explained more clearly
>
> We explain the motivation behind our metrics in **Section 3** and show how to use them in **Sections 5.1 and 5.2**; in the introduction of the paper, we have tried to outline the need for hypernymy metrics to the best of our ability. If you have **specific questions or suggestions** that would help the reader understand ISP and SCS better, we would be happy to follow them.

---

> > ### Comment · Reviewer_4BDw · 2023-11-22
> >
> > Thanks for the response! I do not have further questions and keep the rating unchanged.

---

### Official Review · Reviewer_pgJu · 2023-10-30

**Soundness:** 3 good
**Presentation:** 3 good
**Contribution:** 3 good
**Rating:** 6
**Confidence:** 4

**Summary:**

This paper presents metrics, In-Subtree Probability (ISP) and Subtree Coverage Score (SCS), based on WordNet and ImageNet, for evaluating the hypernymy comprehension capabilities of popular text-to-image models. The proposed method also can be used to provide insights into text-to-image models' limitations for downstream applications. The authors evaluate publicly available models and analyze the hypernymy understanding of existing text-to-image models, validating the effectiveness of their proposed method in this work.

**Strengths:**

1. This study investigates the problem of assessing the hypernymy understanding of text-to-image models. The authors propose an effective approach that combines the hypernymy knowledge from WordNet with the classification capabilities of established image classifiers. This integration demonstrates a successful application in evaluating the hypernymy understanding of the models.

2. The paper was, in general, easy to follow. In particular, I feel condent that, based on the description, I can reimplement the model and reproduce the results.

3. The proposed evaluation framework and its motivation are reasonable (but see the weakness).

**Weaknesses:**

1. The efficacy of the proposed method presented in this study is heavily dependent on the performance of the image classifier. Both the In-Subtree Probability (ISP) and Subtree Coverage Score (SCS) metrics rely on the probabilities generated by the classifier. To achieve reliable results, it is crucial to obtain a highly accurate classifier capable of effectively covering a wide range of real-world object classes. The metrics' effectiveness is contingent on the classifier's performance, meaning that if the classifier demonstrates low prediction accuracy, the reliability of the ISP and SCS metrics may be compromised.

2. Pretrained generic large language models (e.g., T5, Llama2), trained on text-only corpora, demonstrate proficiency in text encoding for image synthesis. These models inherently possess semantic understanding and effectively acquire knowledge of hypernymy. More experiments of applying the proposed metrics into text-to-image models equipped with LLMs need to be conducted.

**Questions:**

1. Please consider extending the experiments with more LLMs-equiped text-to-image models, e.g. Imagen (Photorealistic Text-to-Image Diffusion Models with Deep Language Understanding) (but not limited to.).

---

> ### Author Response · Authors · 2023-11-20
> **Author Response to Official Review by Reviewer pgJu**
>
> Thank you for your positive feedback and insightful comments! Our responses to your questions and concerns can be found below:
>
> > The efficacy of the proposed method presented in this study is heavily dependent on the performance of the image classifier.
>
> We agree, which is why **we use an accurate and robust pretrained ImageNet classifier** (ViT-B/16) in our evaluation and in the reference implementation of the metrics we propose. For a detailed discussion about the validity of using pretrained classifiers, please see the **general response**; overall, we argue that this dependence should not be viewed as a disadvantage of our approach.
>
> > Pretrained generic large language models (e.g., T5, Llama2), trained on text-only corpora, demonstrate proficiency in text encoding for image synthesis
> > Please consider extending the experiments with more LLMs-equiped text-to-image models, e.g. Imagen
>
> Thank you for this suggestion! We would like to note that **Imagen is not publicly available**; therefore, it is not possible to run experiments on this model. Moreover, large language models **do not necessarily have much better hypernymy knowledge**: as shown in [1], models like ChatGPT can fall behind BERT, CLIP, or other smaller neural networks in terms of hypernym discovery, and GPT embeddings are only marginally better when compared to CLIP (which is used in the majority of the models that we study).
>
> Still, to address your concern, we conducted experiments on DeepFloyd IF XL (which has **T5-XXL as its backbone**) and included results in the updated revision of the paper. We kindly ask you to see the new versions of **Tables 1, 9, and 10** and tell us if these results address your concern. In the camera-ready version, we will improve the fidelity metrics of DeepFloyd with super-resolution (that was harder to run within the limits of the response period), but other results should remain similar.
>
> [1] Concept Understanding in Large Language Models: An Empirical Study. Liao et al., ICLR 2023 Tiny Papers track.

---

> > ### Comment · Reviewer_pgJu · 2023-11-22
> >
> > Thanks for the response! Glad to see the experimental results of those models equipped with LLM. I have no further questions and keep the rating unchanged.

---

### Official Review · Reviewer_npm8 · 2023-11-02

**Soundness:** 3 good
**Presentation:** 3 good
**Contribution:** 3 good
**Rating:** 6
**Confidence:** 3

**Summary:**

This paper proposes a framework to evaluate the hypernymy understanding abilities of text-to-image models, including two distinct and complementary metrics. The evaluation process is fully automated. The authors also show some merits of the evaluation framework such as finding unknown concepts and conducting granular comparison of models. Overall, this paper presents a focused analysis on a specific aspect of text-to-image generation (i.e., hypernymy understanding).

**Strengths:**

- This experiments presented in this paper are comprehensive and elaborate.
- The evaluation metrics are automated and could be useful for future research.

**Weaknesses:**

- This paper focuses on a very specific aspect of text-to-image generation.
- The proposed evaluation framework relies on a well-trained image classifier. It performance depends on the accuracy and coverage of the image classifier. Particularly, its usefulness may be limited by the coverage of existing image classifiers.
- The results of some experiments  (e.g., the influence of the classifier-free guidance scale, the number of diffusion steps, and the number of generated samples) seem to be self-evident and provide little new insights.

**Questions:**

- For Subtree Coverage Score, why not use simpler formula such as the entropy of the average distribution.
- Do you anticipate any more applications of the proposed evaluation framework?

---

> ### Author Response · Authors · 2023-11-20
> **Author Response to Official Review by Reviewer npm8**
>
> Thank you for taking the time to review our work and for your feedback! Please find our answers to your concerns below:
>
> > This paper focuses on a very specific aspect of text-to-image generation.
>
> While we agree that we study a particular aspect of generation abilities, we believe that the **hypernymy relation is a sufficiently abstract concept** that can be broadly useful. As we notice in the paper, there are multiple applications of hypernymy that encompass different questions faced by text-to-image generation practitioners. Moreover, as shown by prior work [1], **even specific aspects could be of significant interest** to the community. Our metrics offer a general foundation for a more semantics-driven approach to evaluation, which makes it possible for the research community to explore nuances not captured by existing methods.
>
> > The proposed evaluation framework relies on a well-trained image classifier. It performance depends on the accuracy and coverage of the image classifier. Particularly, its usefulness may be limited by the coverage of existing image classifiers.
>
> Thank you for this observation! We believe that the existence of many **high-quality and robust pretrained ImageNet classifiers** makes our approach viable; for more details, see our **general response**.
>
> > The results of some experiments (e.g., the influence of the classifier-free guidance scale, the number of diffusion steps, and the number of generated samples) seem to be self-evident and provide little new insights.
>
> **These experiments were designed to have unsurprising results:** their goal is to show that our metrics **behave as expected** when sampling hyperparameters are changed and that the default hyperparameters we use for evaluation provide sensible results. In other words, these experiments are **necessary to validate the design choices** of our evaluation protocol, which, in our opinion, should be helpful for intended users of ISP and SCS.
>
> > For Subtree Coverage Score, why not use simpler formula such as the entropy of the average distribution.
>
> Thank you for this question! In general, our motivation is the same as in the Inception Score paper: using the entropy of the average distribution **does not allow us to distinguish** between diverse images of specific objects and images covering mixed objects in one sample. More concretely, we have measured the entropy of the average distribution for setups from Section 4.3 and added its correlation with human preferences to **Appendix H**. As you can see there, measuring entropy in each synset has lower agreement than SCS.
>
> > Do you anticipate any more applications of the proposed evaluation framework?
>
> Apart from **3 applications that we already show in the paper** (finding unknown/low-diversity concepts, conducting pairwise comparisons, and running per-subtree evaluation), we think it is also possible to leverage the hypernymy relation in the following ways:
>
> 1. We can derive a new measure of similarity between two synsets by dividing the total probability of the intersection of classifiable subtrees by the probability of their union. Intuitively, “canine” and “dog” should be considered similar, whereas “animal” and “dog” will be less similar due to “animal” containing additional concept subtrees.
> 2. Similarly, one can compare how two models “view” the same synset by considering its classifiable subtree and computing the Jensen-Shannon divergence (or any other symmetric distribution similarity measure) between hyponym distributions induced by these models.
> 3. We are able to evaluate the multilingual capabilities of models out of the box by employing multilingual WordNet variations [2].
>
> [1] Benchmark for Compositional Text-to-Image Synthesis. Park et al., NeurIPS 2021 Datasets and Benchmarks
>
> [2] Open Multilingual WordNet. https://omwn.org/

---

> ### Comment · Reviewer_npm8 · 2023-11-22
>
> Thank you for the thoughtful response.
>
> I raise my score from 5 to 6. I would like the ACs to know that (1) overall, I think this paper is a solid work and (2) I just doubt its impact on our research community.

---

### Author Response · Authors · 2023-11-20
**General Response to Reviewers**

We thank all reviewers for encouraging feedback and insightful comments on our work. In particular, we are glad the reviewers appreciated the originality of our approach (**pgJu, 4BDw**), its usefulness for future research (**npm8, 4BDw**), as well as rigorous and comprehensive experiments (**npm8, 4BDw**).

Apart from individual comments on the paper, the reviewers expressed two shared concerns about the use of pretrained image classifiers and the breadth of evaluated models. Below, we address both of these comments.

**Reliance on image classifier performance (npm8, pgJu):** we agree that the metrics we design heavily depend on it. However, **this property is not unique to ISP and SCS**: as far as we know, standard automatic metrics for image generation quality (such as Inception Score and FID) also use pretrained classifiers and yield informative quality measures. Recent works [1,2] propose using vision-language models such as CLIP for evaluation in more nuanced setups; while it is possible to extend our approach (**which we already acknowledged** while discussing limitations in the conclusion) to use multimodal models too, we think the results with standard image classifiers provide **sufficiently informative results while being easier to use and to understand** for the reader. Lastly, the concern about the coverage and performance of modern ImageNet classifiers can be addressed by existing empirical results: Vision Transformers (which we use in our work) have classification quality on the ImageNet test set exceeding 84% and are remarkably **robust to domain shifts** and image perturbations [3]. In summary, we believe that using high-quality pretrained ImageNet classifiers is a natural choice for our setup, and there are no clearly superior alternatives.

**Benchmarking additional models (pgJu, 4BDw):** we have uploaded an updated revision of the work with preliminary results for three new text-to-image synthesis models [4,5,6]. These models represent a more diverse set of architectures and show state-of-the-art generation results among open-access models at the time of writing: in particular, DeepFloyd IF uses a large language model (T5-XXL) for text encoding, which fits the category suggested by **Reviewer pgJu**. Due to limited response time and computational constraints, we generated samples from DeepFloyd without the super-resolution step, which is the likely cause for its poor FID score. Nevertheless, we can see from the **updated Table 1** that all three models rank close to each other in terms of averaged quality metrics (the ISP difference between Kandinsky and SDXL is actually less than 0.001) but show vastly different performance on specific WordNet subtrees (**Tables 9 and 10**).

[1] CLIPScore: A Reference-free Evaluation Metric for Image Captioning. Hessel et al., EMNLP 2021

[2] Benchmark for Compositional Text-to-Image Synthesis. Park et al., NeurIPS 2021 Datasets and Benchmarks

[3] Intriguing Properties of Vision Transformers. Naseer et al., NeurIPS 2021.

[4] SDXL: Improving Latent Diffusion Models for High-Resolution Image Synthesis. Podel et al., 2023

[5] Kandinsky: an Improved Text-to-Image Synthesis with Image Prior and Latent Diffusion. Razzhigaev et al., 2023

[6] DeepFloyd IF. https://github.com/deep-floyd/IF

---

### Meta-Review · Area_Chair_wwvW · 2023-12-05

**Metareview:**

The paper proposes a heuristic metric to evaluate the hypernymy (i.e., is-a relation) understanding of text-to-image models. The experiments were conducted with a few text-to-image models (such as GLIDE, Latent Diffusion, and Stable Diffusion) with consideration of several key hyperparameters.

Overall, this is a borderline paper. The reviewers generally agree that the paper is well written, and that the approach is reasonable and may be useful.

However, a few major concerns are also raised.

The proposed approach also heavily depends on an underlying image classifier, which may affect the efficacy of the proposed metric. The paper appears narrow in its scope, yet the approach is heuristic (without a solid foundation). I would encourage the authors to further improve their work and consider more focused publication venues.

**Justification For Why Not Higher Score:**

The paper appears narrow in its scope, yet the approach is heuristic and also ad hoc to the setting. The approach highly depends on an underlying image classifier.

**Justification For Why Not Lower Score:**

N/A

---

### Decision · Program_Chairs · 2024-01-16

Reject